# Freedom to choose between public resources promotes cooperation

**Mohammad Salahshour** *

Max Planck Institute for Mathematics in the Sciences, Leipzig, Germany

* mohammad.salahshour@mis.mpg.de

## Abstract

As cooperation incurs a cost to the cooperator for others to benefit, its evolution seems to contradict natural selection. How evolution has resolved this obstacle has been among the most intensely studied questions in evolutionary theory in recent decades. Here, we show that having a choice between different public resources provides a simple mechanism for cooperation to flourish. Such a mechanism can be at work in many biological or social contexts where individuals can form different groups or join different institutions to perform a collective action task, or when they can choose between collective actions with different profitability. As a simple evolutionary model suggests, defectors tend to join the highest quality resource in such a context. This allows cooperators to survive and out-compete defectors by sheltering in a lower quality resource. Cooperation is maximized, however, when the qualities of the two highest quality resources are similar, and thus, they are almost interchangeable.

**Data Availability Statement:** All the Matlab data and Matlab codes used in this study are publicly available at https://doi.org/10.5281/zenodo.4294725. Matlab codes are presented in the Supplementary information.

## Author summary

How is selfishness curbed in many biological populations to solve collective action problems? A simple answer to this question arises by noting that, in many populations, individuals can join different groups to perform a collective action task. Similarly, individuals often face situations where they can join different collective action tasks with different profitability. As we show here, this provides a natural mechanism for solving collective action problems. In such contexts, defectors predominantly join the higher quality public resource. This enables cooperators to survive and out-compete defectors by working cooperatively in a lower quality resource. However, cooperation is maximized when different public resources have similar qualities, and thus, similar attractiveness for the individuals.

## Introduction

The public goods game, also known as $n$ person prisoner's dilemma, has been one of the paradigms used in many studies on the evolution of cooperation [1–21]. This game is played among $n$ individuals. Each individual can either cooperate or defect. Cooperators contribute an amount $c$ to a public good. Defectors contribute nothing. All the contributions are multiplied by an enhancement factor $r < n$ and are distributed equally among the players. When

**Funding:** The author acknowledges funding from Alexander von Humboldt Foundation in the framework of the Sofja Kovalevskaja Award endowed by the German Federal Ministry of Education and Research. The funders had no role in study design, data collection and analysis, decision to publish, or preparation of the manuscript.

**Competing interests:** The authors have declared that no competing interests exist.

played in an evolutionary context in a well-mixed population, defectors, not paying the cost of cooperation, reach the highest payoff and dominate the population. However, contrary to this expectation, empirical evidence from bacteria to animals and humans suggests a high level of cooperation has evolved in the course of evolution [22–25]. As a result of many attempts to resolve this apparent paradox, some mechanisms which are capable of promoting cooperation in public goods game have come into the light [26–29]. Examples include indirect reciprocity, such as reputation effects [1, 2], direct reciprocity [3, 4], network and spatial structure [5–7], voluntary participation [7–9], reward [10–14], and punishment [15–21].

As we show here, freedom to choose between different public resources provides yet another road to the evolution of cooperation. When having a choice between multiple public resources, individuals can form sub-groups of varying sizes. In sufficiently small sub-groups, cooperation becomes dominant. This leads to the evolution of cooperation in the population. This mechanism can be at work in a situation where individuals can choose between different public resources to perform a collective action task. Such a situation can naturally occur in many examples of collective action dilemmas observed in animal and human societies [30]. For instance, different public resources can be different collective actions with different profitability, different groups an individual can join to perform a collective action task, such as fishing or foraging [30, 31], or different places or groups an individual can join to perform an economic activity [32–35]. As our analysis shows, in such a situation, defectors tend to join the highest quality public resource. This enables cooperators to survive and out-compete defectors by sheltering in a lower quality public resource. This phenomenon is reminiscent of some arguments according to which cooperative strategies fare better in harsh environments [36–38]. However, cooperation is maximized when the two highest quality public resources have similar qualities, and thus, similar attractiveness for the individuals. Thus, our results show that being able to choose between different collective action tasks provides a natural mechanism for solving collective action problems.

## The model

To see how the freedom to choose between different public resources promotes cooperation, we consider a general framework where $n \geq 2$ public resources with enhancement factors $r_1$ to $r_n$ exist. For simplicity, we begin with a case where two different public resources with enhancement factors $r_1$ and $r_2$ exist. More precisely, we consider a population of $N$ individuals, each having a strategy $s$ (which can be cooperation ($C$), or defection ($D$)), and a preferred game $i$ (which can be game 1 or game 2). At each time step, groups of $g$ individuals are formed at random from the population pool. Individuals in each group enter their preferred public goods game, play the game, and derive payoffs according to the outcome of the game. In addition, they receive a base payoff $\pi_0$ from other activities not related to the public goods game. After playing the games, individuals reproduce with a probability proportional to their payoff. The new generation replaces the old one such that the population size remains constant. Offspring inherit the strategy and the preferred game of their parents, subject to mutations. Mutations in the strategy and the preferred game occur independently, each with a probability $v$. In the case that a mutation occurs, the value of the corresponding variable is flipped to its opposite value (e.g., $C$ to $D$ for the strategy and game 1 to game 2 for the preferred game).

## Results

### Dynamics of the model and the phase diagram for $n = 2$ public resources

As shown in the section Materials and methods, the model can be solved analytically in terms of the replicator-mutator equation. Analysis of the model by simulations and numerical

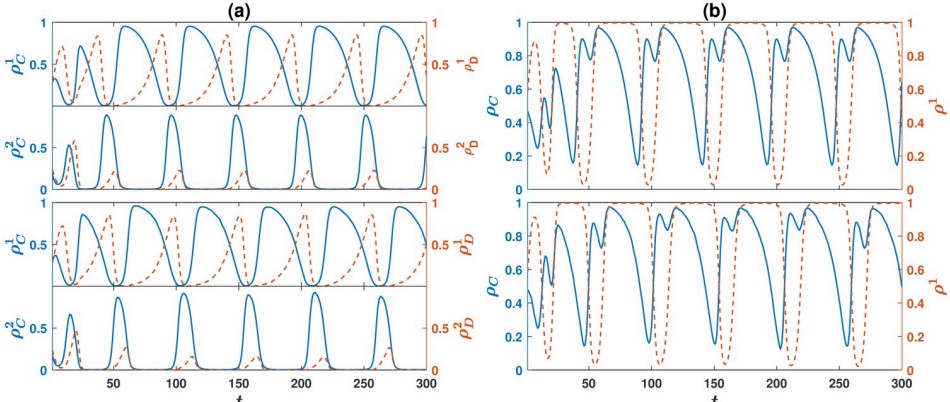

**Fig 1. Dynamics of the model for $n$ = 2 public resources.** A: The density of cooperators (solid blue), $\rho_C^i$, and defectors (dashed red), $\rho_D^i$, in a game $i$, for $i$ = 1 and 2, as a function of time. The two top panels result from the replicator dynamics, and the two bottom panels result from a simulation. B: The total density of cooperators, $\rho_C = \rho_C^1 + \rho_C^2$ (solid blue), and the density of the individuals who prefer public resource 1, $\rho^1 = \rho_C^1 + \rho_D^1$ (dashed red), as a function of time. The top panel results from replicator dynamics and, the bottom panel results from a simulation. Parameter values: $g$ = 10, $v = 10^{-3}$, $r_1$ = 4.4, and $r_2$ = 2.8. Simulations are performed in a population of size $N$ = 10000.

solution of the replicator dynamics reveals that the dynamics can settle in a fixed point or a periodic orbit depending on the parameter values and the initial conditions. We begin by studying cyclic behavior. In Fig 1A, we plot the density of cooperators (solid blue) and defectors (dashed red) in the game $i$, denoted as, respectively, $\rho_C^i$ and $\rho_D^i$, for $i$ = 1 and $i$ = 2, as a function of time. The density of cooperators (defectors) in the game $i$ is defined as the fraction of individuals in the population who prefer public resource $i$ and cooperate (defect). In Fig 1B the total density of cooperators $\rho_C = \rho_C^1 + \rho_C^2$ (solid blue), and the density of the individuals who prefer public resource 1, $\rho^1 = \rho_C^1 + \rho_D^1$, (dashed red) are plotted. The total density of defectors, $\rho_D = \rho_D^1 + \rho_D^2$, and the density of individuals who prefer public resource 2, $\rho^2 = \rho_C^2 + \rho_D^2$, can be inferred from these by noting $\rho_D = 1 - \rho_C$, and $\rho^2 = 1 - \rho^1$. Here, $g$ = 10, $r_1$ = 4.4, $r_2$ = 2.8, $v = 10^{-3}$, and the initial condition is a random initial condition in which both the strategy and the preferred game of the individuals are assigned at random. Here, and in the following, we fix $c = \pi_0 = 1$. The top panels show numerical solutions of the replicator dynamics, and the bottom panels show the results of a simulation in a population of size $N$ = 10000.

A comparison shows a high level of agreement between simulations in a finite population and solutions of replicator dynamics, which is an exact solution of the model in the infinite population limit. Both simulation and analytical solutions show that cooperators are preserved and cyclically dominate the population. When cooperators' density in a public resource $i$ increases, defectors in the game $i$ obtain a high payoff and increase in frequency. This decreases the payoff of cooperators in the game $i$, and consequently, they obtain a higher payoff by switching to the other public resource. This, in turn, decreases the payoff of defectors in the game $i$, and thus $\rho_D^i$ starts to decrease when $\rho_C^i$ decreases enough.

As mentioned before, cyclic behavior is not the only attractor of the dynamics. To see this, in Fig 2A and 2B, the phase diagrams of the system in the $r_1 - r_2$ plane for $g$ = 10, and respectively, $v = 10^{-5}$ and $v = 10^{-3}$ are plotted. For too small enhancement factors, having a choice between public resources can not alleviate the strong disadvantage cooperators experience due to a small enhancement factor. Consequently, the dynamics settle into a fixed point in which defectors dominate the population. As either enhancement factors increase beyond a threshold (marked by blue triangles), cooperators can survive. In this region, the

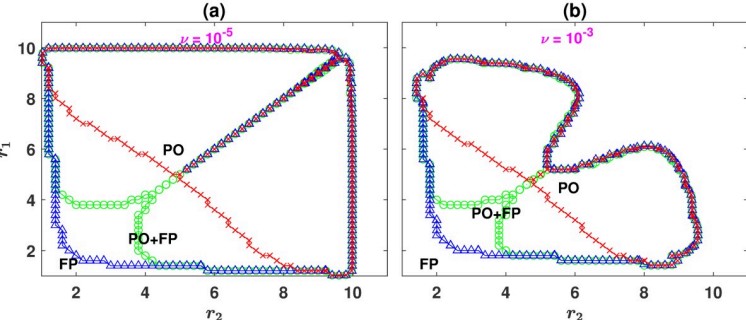

**Fig 2. The phase diagram of the model for $n$ = 2 public resources.** The phase diagram of the model for $v = 10^{-5}$, A, and $v = 10^{-3}$, B. The model shows two distinct mono-stable phases where the dynamics settle into a fixed point (denoted by FP) or a periodic orbit (denoted by PO). In between, for small enhancement factors, the system possesses a bi-stable region where both the periodic orbit and the fixed point are stable (the region between blue (triangles) and red (crosses) lines, denoted by PO+FP). The phase boundary (green line marked by circles) can be defined as the line where the transition between the two phases occurs starting from a random assignment of strategies (that is $\rho_C^1 = \rho_C^2 = \rho_D^1 = \rho_D^2 = 0.25$). Here, $g$ = 10 and the replicator dynamics is used to derive the phase diagrams.

dynamics become bistable, and depending on the initial condition, settle either in a defective fixed point or a periodic orbit. By further increasing the enhancement factors, at a second bifurcation line (denoted by red crosses), the defective fixed point loses stability. The dynamics become mono-stable and settle in a periodic orbit, starting from all the initial conditions. The phase boundary between the defective and cyclic phases in the $r_1 − r_2$ plane can be defined as the line where the transition between the two phases occurs starting from a homogeneous initial condition (that is $\rho_C^1 = \rho_D^1 = \rho_C^2 = \rho_D^2 = 0.25$) [39]. Green circles in the figure denote this boundary.

By further increasing the enhancement factors, beyond a final transition line (the line where different marker types coalesce), the periodic orbit becomes unstable, and the dynamics settle in a fixed point in which cooperators dominate. We note that in contrast to the transition from the defective fixed point to the cyclic orbit, the transition between the cyclic orbit and the cooperative fixed point shows no bi-stability and happens continuously (in the sense that the amplitude of fluctuations decreases gradually by increasing the enhancement factors until fluctuations vanish at the transition line). Consequently, this transition line occurs at the same value of $r_1$ and $r_2$ for all initial conditions. A comparison of the phase diagram for different mutation rates shows that the same scenario occurs for all the mutation rates. Quantitatively, however, larger mutation rates decrease the size of the region in the $r_1 − r_2$ plane where cyclic behavior can occur.

## Cooperation level of the population for $n$ = 2 public resources

An interesting question is how the level of cooperation in the population depends on the enhancement factors? To see this, in Fig 3A, we plot the time average of the total density of cooperators, $\rho_C = \rho_C^1 + \rho_C^2$, in the $r_1 − r_2$ plane. In all panels of Fig 3, $g$ = 10, $v = 10^{-3}$, and a homogeneous initial condition is used. The top panels result from the replicator dynamics, and the bottom panels show the result of a simulation in a population of size $N$ = 20000. As can be seen, the density of cooperators is the highest close to the diagonal, where $r_1 = r_2$, and decreases with increasing the distance from the diagonal. This shows that increasing the enhancement factor of one of the games beyond its optimal value can have a surprisingly detrimental effect on the evolution of cooperation.

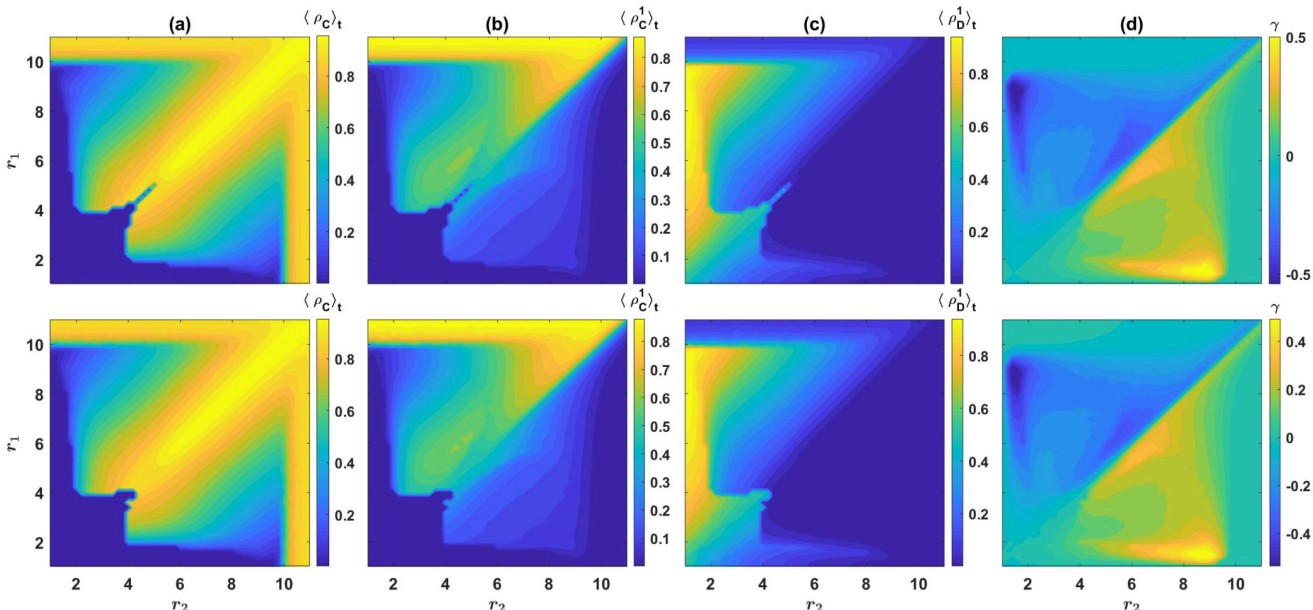

**Fig 3. Cooperation level of the population in the model with $n = 2$ public resources.** A: The contour plot of the time average density of cooperators, $\langle \rho_C \rangle_t = \langle \rho_C^2 + \rho_C^1 \rangle_t$. Cooperation evolves for large enough enhancement factors, and it is maximized when the enhancement factors of the two resources are similar. B and C: The time average density of cooperators B, and defectors C, in resource 1. Compared to cooperators, defectors are more likely to prefer the highest quality resource. Consequently, cooperators can use the low-quality resource as a shelter when defectors dominate. D: Contour plot of $\gamma = \langle \rho_C^1/(\rho_C^1 + \rho_C^2) - \rho_D^1/(\rho_D^1 + \rho_D^2) \rangle_t$. This is a measure of how much the cooperators are more likely to prefer resource 1, compared to defectors. Cooperators are more likely to prefer resource 1, when it is the lower quality resource, and defectors are more likely to prefer resource 1 when it is the higher quality resource. Here, $g = 10$ and $v = 10^{-3}$. Top panels result from the replicator dynamics, and bottom panels show the result of a simulation in a population of size $N = 20000$.

To shed more light on the mechanism by which freedom to choose between different public resources promotes cooperation, and to see why cooperation is maximized when the two competing resources have similar enhancement factors, in Fig 3B and 3C, we plot the time average density of, respectively, cooperators and defectors in resource 1 (Due to the symmetry of the two games, the corresponding densities in resource 2 result from this by reflection with respect to the diagonal). As can be seen, most individuals prefer resource 1, as long as it has a higher enhancement factor. Interestingly, the low-quality resource is often immune to invasion by defectors: while the inferior resource can attract cooperators, defectors strongly prefer the higher quality resource. This phenomenon can be made more quantitative by calculating the fraction of cooperators who prefer resource 1 as $\rho_C^1/(\rho_C^1 + \rho_C^2)$, and the fraction of defectors who prefer resource 1, $\rho_D^1/(\rho_D^1 + \rho_D^2)$. These quantities measure how likely a cooperator or a defector is to prefer resource 1. The time average difference between these two quantities, $\gamma = \langle \rho_C^1/(\rho_C^1 + \rho_C^2) - \rho_D^1/(\rho_D^1 + \rho_D^2) \rangle_t$ is plotted in Fig 3D. As can be seen, cooperators are always more likely to prefer the lower quality resource than the defectors are. On the other hand, defectors are more likely to prefer the higher quality resource than the cooperators are. This phenomenon is crucial for freedom to choose between the two resources to promote cooperation: When the density of defectors increases in the population, cooperators can shelter in the lower quality resource and make it the more profitable one, and thus, by enjoying a higher growth rate, out-compete defectors. However, when the inferior resource's quality is much lower than that of the superior resource, it becomes more difficult for cooperators to make the inferior resource the more profitable one. This, in turn, leads to a reduction in the cooperation level of the population by increasing the difference between the quality of the resources.

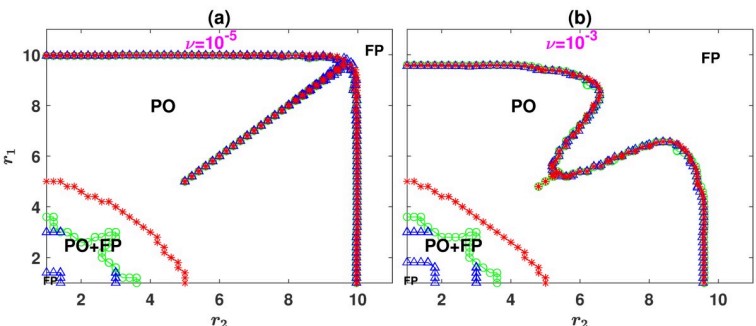

**Fig 4. The phase diagram of the model with $n = 3$ public resources.** The phase diagram of the model with $n = 3$ public resources, for $v = 10^{-5}$, A, and $v = 10^{-3}$, B. For small enhancement factors, the model shows bistability and can settle in either a fixed point (FP) where defectors dominate or a periodic orbit (PO) in which cooperators survive. The blue line (marked by triangles) shows the lower boundary of the coexistence region above which the cyclic orbit becomes stable, and the red line (marked by crosses) shows the upper boundary of the coexistence region, above which the defective fixed point becomes unstable. The green line (marked by circles) shows the phase boundary resulted from an unbiased (homogeneous) initial condition. Here, $g = 10$, $r_3 = 3$, and the replicator dynamics are used to derive the phase diagrams.

## The case of $n = 3$ public resources

As mentioned in the section The model, the framework can be extended to a general case where individuals can choose among an arbitrary number, $n$, of public resources. In this section, we study the case of $n = 3$ public resources. The phase diagram of the model with three resources is plotted in Fig 4A and 4B, for respectively, $v = 10^{-5}$ and $v = 10^{-3}$. Here, the replicator dynamics is used to derive the phase diagrams in the $r_1 - r_2$ plane, fixing the enhancement factor of game 3 to $r_3 = 3$, and setting $g = 10$. The model shows a similar phase diagram to that observed for the case of $n = 2$ resources: For small enhancement factors, the dynamics settle into a defective fixed point. As the enhancement factors increase, the system becomes bistable and can settle in either the defective fixed point or a periodic orbit in which cooperators survive and cyclically dominate the population. The blue line (marked by blue triangles) shows the lower boundary of the coexistence region above which the periodic orbit becomes stable. The red line (marked by crosses) shows the upper boundary of the coexistence region, above which the defective fixed point becomes unstable, and the dynamics settle in the periodic orbit. In between the two bifurcation lines and in the coexistence region, both the defective fixed point and the periodic orbit are stable, and the dynamics settle into one of these attractors depending on the initial conditions. The phase boundary, defined as the boundary above which the transition from the fixed point to periodic orbit occurs starting from a homogeneous initial condition [39], is indicated by the green line (marked by circles). Finally, for large enhancement factors, the system shows a final bifurcation line above which the dynamics settle in a cooperative fixed point, in which the vast majority of the individuals cooperate. This boundary is indicated by the line in large enhancement factors where different marker types coincide.

To see how the cooperation level of the population changes in different regions of the phase diagram, in Fig 5A and 5B, we plot the time average population's cooperation level, $\langle \rho_C \rangle_t = \langle \rho_C^1 + \rho_C^2 + \rho_C^3 \rangle_t$. Here, and in Fig 6 in the following, $v = 10^{-3}$, $g = 10$, and $r_3 = 3$. The initial condition is a homogeneous initial condition in which the individuals' strategies and preferred games are randomly assigned. Fig 5A shows the solutions of the replicator dynamics, and Fig 5B shows the result of simulations in a population of $N = 10000$ individuals. As was the case in the model with $n = 2$ resources, the cooperation level of the population is

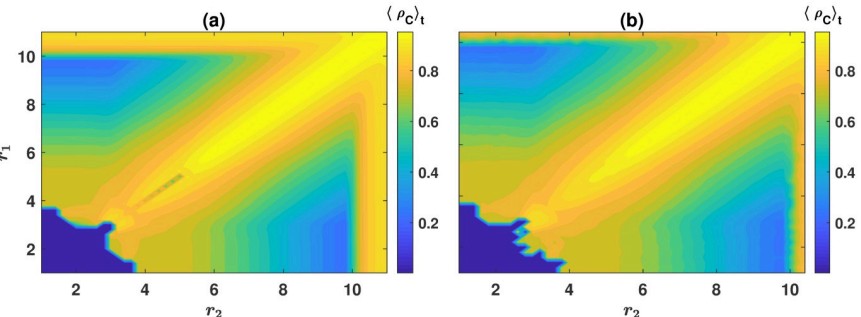

**Fig 5. Cooperation level of the population in the model with $n = 3$ public resources.** The contour plots of the time average cooperation level of the population, $\langle \rho_C \rangle_t = \langle \rho_C^1 + \rho_C^2 + \rho_C^3 \rangle_t$, resulting from numerical solutions of the replicator dynamics with $n = 3$ games, A, and a simulation in population of size $N = 10000$, B. The population's cooperation level is the highest when the two highest quality resources have similar enhancement factors. Here, $g = 10$, $r_3 = 3$, and $v = 10^{-3}$. The initial condition is a homogeneous initial condition in which the strategies and the preferred game of the individuals are assigned at random.

maximized close to the diagonal, where the two higher quality games have similar enhancement factors and decreases by going away from the diagonal.

To take a deeper look into the model's behavior for $n = 3$ games, we plot the time average density of cooperators in resource 1, $\langle \rho_C^1 \rangle_t$, and that of defectors in resource 1, $\langle \rho_D^1 \rangle_t$, in respectively, Fig 6A and 6B. The time average density of cooperators in resource 3, $\langle \rho_C^3 \rangle_t$, and the time average density of defectors in resource 3, $\langle \rho_D^3 \rangle_t$ are plotted in, respectively, Fig 6C and 6D. The top panels show the result of the replicator dynamics, and the bottom panels show the result of simulations in a population of size $N = 10000$. We note that, due to the symmetry of

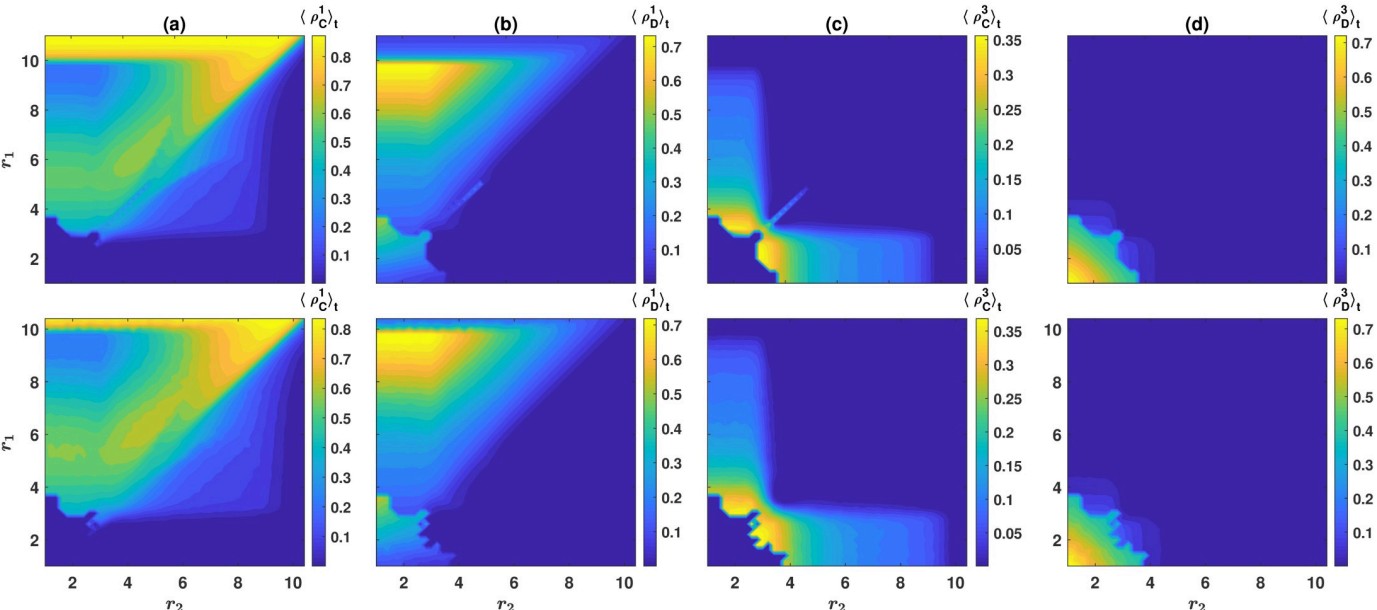

**Fig 6. The densities of different strategies in the model with $n = 3$ public resources.** The time average density of cooperators in resource 1 A, the time average density of defectors in resource 1 B, the time average density of cooperators in resource 3 C, and the time average density of defectors in resource 3 D, are plotted. The top panels show the replicator dynamics results, and the bottom panels show the result of a simulation in a population of size $N = 10000$. Here, $g = 10$, $r_3 = 3$, and $v = 10^{-3}$. The initial condition is a homogeneous initial condition in which the strategies and the preferred games of the individuals are assigned at random.

resource 1 and 2, the densities of cooperators and defectors in resource 2 result from that in resource 1, in Fig 6, by reflection with respect to the diagonal.

In the region where both $r_1$ and $r_2$ are larger than $r_3 = 3$, that is when resource 3 is the most inferior resource, the third resource becomes almost obsolete: Very few individuals prefer resource 3, and the dynamic is determined by the interaction of the two higher-quality resources. However, the mere possibility of playing the third game has a positive effect on cooperation, even when it does not attract individuals significantly. We note that, in the case $r_1 = r_2$, it can happen that cooperators, but not defectors, choose resource 3, provided its enhancement factor is not too small compared to the enhancement factors of the two higher-quality resources. This, once again, shows cooperators are more likely to choose an inferior resource.

On the other hand, in the region where either $r_1$ or $r_2$ are smaller than $r_3 = 3$, that is when resource 3 is the second high-quality resource, the third resource always attracts cooperators, but not defectors (compare Fig 6C and 6D). This observation shows, similarly to the case of $n = 2$ resources, when individuals can choose between three resources, defectors tend to prefer the highest quality resource. This makes a second high-quality resource a shelter for cooperators to work cooperatively and grow safely when the density of defectors increases in the population. The lowest quality resource, on the other hand, remains almost obsolete (as long as its quality is sufficiently lower than the other two resources). This is because the second high-quality resource, having a low fraction of defectors, is already a good enough shelter for cooperators, and they do not need to switch to the lowest quality resource.

## Discussion

We have shown that freedom to choose between different public resources can promote cooperation. In such competitive contexts, while defectors predominantly prefer the highest quality resource, cooperators show an inclination to choose lower-quality resources as well, where they can work cooperatively to out-compete defectors. However, the cooperation level of the population is maximized when the two highest quality resources have similar qualities. In the Supporting Information S1 Text, we further argue for the robustness of our result for other parameter values. In particular, finite-size analysis of the model reveals, while having a choice between different public resources promotes cooperation in small population sizes, this mechanism is more potent in large population sizes [See Figure B in S1 Text]. In this regards, having a choice between public resources seems to contrast many mechanisms for the evolution of cooperation which suffer from the so-called scalable cooperation problem [40, 41], according to which cooperation diminishes in large population sizes. Our analysis thus shows freedom to choose between public resources can fruitfully be at work to give rise to a high level of cooperation not only in small scale societies but more prominently, in many large scale societies where individuals can choose among different groups or activities, possibly with different profitability, to engage in collective actions.

Empirical examples of a situation where having a choice between public resources is at work to promote cooperation appear to abound in different species, from resource competition in microorganisms [37, 42] to many collective actions in human societies [32–35, 43]. For instance, an interesting example in microorganisms appears to be resource competition in yeast populations, where individual organisms can choose between different resources with different profitability [37].

Fission-fusion dynamics, observed in many group-living animals [44, 45], such as primates [46–48], dolphins [49], elephants [50], bats [51], carnivores [52–54], and hunter-gatherer

human societies [44, 55], appears to provide another example. In such cases, subgroups are formed in the group, for instance, during foraging [49, 53, 55], cooperative breeding [48], or cooperative defending [53]. The composition of subgroups are usually dynamic, and individuals can choose or form different subgroups or switch between subgroups in a matter of hours (for example, in spotted hyenas [53]) to a day or a few months (for example, in hunter-gatherer human societies [55]). In many such cases, individuals in subgroups need to work together to solve collective action problems. Our analysis suggests the fission-fusion dynamics can serve as a natural mechanism to solve such collective action problems. This role in solving collective action problems, in turn, can be argued to be one of the advantages of fission-fusion dynamics. Although the existence of possible links between cooperative behavior and fission-fusion dynamics has been noted [48, 51], the question of whether fission-fusion dynamics serve a role in promoting cooperative behavior in many species that exhibit both fission-fusion dynamics and a high level of cooperation remains to be empirically studied.

In modern human societies, competition between different political parties, or team incentive within firms [43], where the success of a group in competition with others require cooperation between individuals, seems to provide examples where individuals can form and join different groups to perform a collective action task. It is argued that competition between different groups can promote cooperation in such contexts [43]. Arguably, teamwork in other human communities, such as schools, where individuals can form and join different groups to perform a collective action task, provides other examples. Choosing between different geographic locations, such as cities, based on public good provision [56, 57], or having a choice between different type of collective action tasks, such as communal hunting or communal agriculture [58], can be among yet other examples of a situation where having a choice between different public resources can be at work to promote cooperation in human societies.

One might wonder what prevents cooperators from going to extinction in our model? The key to the answer to this question comes from the observation that while the group size, $g$, is fixed, the number of individuals who prefer a public resource $i$ in a group, $n^i$, is subject to fluctuations. When in a group, $n^i$ falls below $r_i$, individuals' in this group no longer face a social dilemma, and cooperation becomes the dominant strategy. In this case, although those who defect in a game $i$ fare better in the group, when averaged over all such groups, the game $i$-cooperators reach a higher payoff compared to game $i$-defectors and increase in number. A similar phenomenon is observed in voluntary participation, where it is noted that it is an instance of Simpson's paradox [8]. This phenomenon seems to be particularly the reason why finite-size effects favor defection in our model, as the likelihood that averaging over groups does not wash away the advantage defectors experience in each group increases with decreasing population size.

The mechanism introduced here has similarities with voluntary participation [8]. In the case of voluntary participation, individuals can abstain from playing a public goods game and resort to a safe income. It is shown that abstaining can lead to a rock-paper-scissor like dynamics through which cooperators, loners (those who abstain from the game), and defectors cyclically dominate the population [8]. Our model can exhibit similar cyclic dominance of different strategies. In the case of two public goods games, four strategies, $C^1$, $D^1$, $C^2$, and $D^2$, can cyclically dominate the population. Similarly, in the case of 3 resources, depending on the parameter values, all or a subset of the 6 possible strategies can alternately dominate the population. However, in our model, instead of having the option of resorting to a safe income, individuals can switch to another public goods game, the income from which can be negative or positive depending on the communities' ability to solve the social dilemma. If the

individuals somehow manage to solve the social dilemma in one public goods game, a positive income from this game is assured, which can induce cooperation in the other coupled games through cyclic dominance of cooperators and defectors. If not, defection prevails in all the public goods games in a defective fixed point. In contrast to voluntary participation, this interdependent dynamics of different social dilemmas introduces bistability: The fate of the dynamics depends on the community's ability to solve the social dilemmas, encoded in the initial conditions.

An interesting question is the effect of population structure on the evolution of cooperation when individuals can choose between different public resources. As is well-known, population structure can promote cooperation [5, 6, 29]. For this reason, one might expect due to the constructive role of network assortativity, having a choice between public resources may promote cooperation in a structured population with more ease and in more hostile conditions, compared to an unstructured population. Furthermore, it might be expected that due to smaller effective group size, having a choice between public resources can increase cooperation beyond what is supported by network reciprocity. Preliminary simulations suggest these expectations are correct. Future research can shed light on these questions and on the spatial patterns emerging when individuals have a choice between different public resources.

## Materials and methods

### The replicator dynamics

In the limit of infinite population size, the dynamics of the model can be described in terms of the replicator-mutator equation [59]. This equation describes the evolution of the densities of different strategies in an evolutionary context where the strategies grow proportional to their fitness and are subject to mutations. In the general case that individuals can choose among $n$ different public resources, the replicator-mutator dynamics for the model can be written as follows:

$$\rho_x^i(t+1) = \sum_{y,j} v_{y,j}^{x,i} \rho_y^j(t) \frac{\pi_y^j(t)}{\sum_{z,l} \rho_z^l(t) \pi_z^l(t)}.$$ (1)

Here, $x$, $y$, and $z$ refer to strategies and can be either $C$ or $D$, and $i$, $j$, and $l$ refer to the public resources numbered from 1 to $n$. $v_{y,j}^{x,i}$ is the mutation rate from a strategy combination that prefers public resource $j$ and plays strategy $y$ to a strategy combination that prefers public resource $i$ and plays strategy $x$. These can be written in terms of mutation rates. In the case of $n = 2$ public resources, these read as follows:

$$\begin{cases} v_{y,j}^{x,i} = 1 - v_s - v_g + v_s v_g, & \text{if} \quad i = j \quad and \quad x = y \\ v_{y,j}^{x,i} = v_s - v_s v_g, & \text{if} \quad i = j \quad and \quad x \neq y \\ v_{y,j}^{x,i} = v_g - v_s v_g & \text{if} \quad i \neq j \quad and \quad x = y \\ v_{y,j}^{x,i} = v_s v_g & \text{if} \quad i \neq j \quad and \quad x \neq y \end{cases}$$ (2)

Where we have assumed that the mutation rate in the strategies is $v_s$ and the mutation rate in the game preference is $v_g$ (through this study, we set $v_s = v_g = v$). In Eq (1), $\pi_y^j$ is the expected payoff of an individual who prefers public resource $j$ and plays strategy $y$. These can be written

as:

$$
\pi_C^j = \sum_{n_D^j=0}^{g-1-n_C^j} \sum_{n_C^j=0}^{g-1} cr_j \frac{1+n_C^j}{1+n_C^j+n_D^j}
$$
$$
(1-\rho_C^j-\rho_D^j)^{g-1-n_C^j-n_D^j} \rho_D^{j\,n_D^j} \rho_C^{j\,n_C^j} \begin{pmatrix} g-1 \\ n_C^j, n_D^j, g-1-n_C^j-n_D^j \end{pmatrix} - c + \pi_0,
$$
$$
\pi_D^j = \sum_{n_D^j=0}^{g-1-n_C^j} \sum_{n_C^j=0}^{g-1} cr_j \frac{n_C^j}{1+n_C^j+n_D^j}
$$
$$
(1-\rho_C^j-\rho_D^j)^{g-1-n_C^j-n_D^j} \rho_D^{j\,n_D^j} \rho_C^{j\,n_C^j} \begin{pmatrix} g-1 \\ n_C^j, n_D^j, g-1-n_C^j-n_D^j \end{pmatrix} + \pi_0.
$$

(3)

Here, $cr_j \frac{1+n_C^j}{1+n_C^j+n_D^j} - c$ in the first equation is the expected payoff of a cooperator who prefers the public resource $j$, in the case that $n_C^j$ cooperators and $n_D^j$ defectors in the group prefer the public resource $j$. Similarly, $cr_j \frac{n_C^j}{1+n_C^j+n_D^j}$ in the second equation is the expected payoff of a defector who prefers the public resource $j$, in the case that $n_C^j$ cooperators and $n_D^j$ defectors in the group prefer the public resource $j$. $(1-\rho_C^j-\rho_D^j)^{g-1-n_C^j-n_D^j} \rho_D^{j\,n_D^j} \rho_C^{j\,n_C^j} \begin{pmatrix} g-1 \\ n_c^j, n_D^j, g-1-n_C^j-n_D^j \end{pmatrix}$ is the probability that this event occurs. $\begin{pmatrix} g-1 \\ n_c^j, n_D^j, g-1-n_C^j-n_D^j \end{pmatrix} = \frac{(g-1)!}{n_C^j! n_D^j! (g-1-n_C^j-n_D^j)!}$ is the multinomial coefficient and is the number of ways that $n_C^j$ cooperators and $n_D^j$ defectors who prefer game $j$ can be chosen among $g-1$ group-mates of a focal individual. Summation over all the possible configurations gives the expected payoff of a cooperator (or defector) who prefers resource $j$. As all the individuals receive a base payoff $\pi_0$, this is added to all the payoffs.

## The simulations and methods

Simulations used in Figs 3, 5, and 6 are run for $T = 5000$ time steps, and the time averages are taken over the last 4000 time steps. For the solutions of the replicator dynamics presented in Figs 3, 5, and 6, the replicator dynamics is numerically solved for $T = 4000$ time steps, and the averages are taken in the time interval $t = 2000$ to $t = 4000$. To derive the phase diagram presented in Figs 2 and 4, the following procedure is used. To determine the boundaries of the bistable region where both defective fixed point and cyclic behavior are possible, we note that the most favorable initial condition for the evolution of cooperation is the one in which all the individuals prefer the same resource (for example, $\rho_D^1 = 1$, $\rho_C^1 = \rho_C^2 = \rho_D^2 = 0$ for $n = 2$ and $\rho_D^1 = 1$, $\rho_C^1 = \rho_C^2 = \rho_D^2 = \rho_C^3 = \rho_D^3 = 0$ for $n = 3$ resources). Numerical solutions of the replicator dynamics starting from this initial condition give the lower boundary of the coexistence region (the blue line marked by triangles in the figures). The least favorable initial condition for the evolution of cooperation is the one in which all the individuals are defectors and prefer the two games in equal fractions ($\rho_C^1 = \rho_C^2 = 0$ and $\rho_D^1 = \rho_D^2 = 0.5$ for $n = 2$ and $\rho_C^1 = \rho_C^2 = \rho_C^3 = 0$ and $\rho_D^1 = \rho_D^2 = \rho_D^3 = 1/3$ for $n = 3$ resources). Numerical solutions of the replicator dynamics starting from this initial condition give the upper boundary of the coexistence region (the red line marked by crosses). Finally, to determine the phase boundary between the defective and cyclic phases (green line marked by circles), the replicator equation with a homogeneous initial condition ($\rho_C^1 = \rho_C^2 = \rho_C^2 = \rho_D^1 = 0.25$ for $n = 2$, and $\rho_C^1 = \rho_C^2 = \rho_C^3 = \rho_D^3 = \rho_D^2 = \rho_D^1 = 1/6$ for $n = 3$ resources) is solved. All the three sets are solved for $T = 4000$ time steps, and the values of $r_1$ and $r_2$ where a periodic orbit appears or disappears mark the phase

boundaries. As the phase boundary between the cyclic and cooperative phases for large enhancement factors shows no bi-stability, solutions starting from all the initial conditions result in the same line where different types of markers coalesce in the figure.

## Supporting information

**S1 Text. Supporting information appendix.** Overview and further analysis of the model, discussion of the dependence on the parameters, and presentation of computer codes.
(PDF)

## Acknowledgments

Fatemeh Bibak is acknowledged for reading the manuscript and comments.

## Author Contributions

**Conceptualization:** Mohammad Salahshour.

**Data curation:** Mohammad Salahshour.

**Formal analysis:** Mohammad Salahshour.

**Funding acquisition:** Mohammad Salahshour.

**Investigation:** Mohammad Salahshour.

**Methodology:** Mohammad Salahshour.

**Project administration:** Mohammad Salahshour.

**Resources:** Mohammad Salahshour.

**Software:** Mohammad Salahshour.

**Supervision:** Mohammad Salahshour.

**Validation:** Mohammad Salahshour.

**Visualization:** Mohammad Salahshour.

**Writing – original draft:** Mohammad Salahshour.

**Writing – review & editing:** Mohammad Salahshour.

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
