## [Decision Letter · Decision Letter 0]

13 Nov 2020

Dear Dr. Salahshour,

Thank you very much for submitting your manuscript "Competition between public resources promotes cooperation" for consideration at PLOS Computational Biology.

As with all papers reviewed by the journal, your manuscript was reviewed by members of the editorial board and by several independent reviewers. In light of the reviews (below this email), we would like to invite the resubmission of a revised version that takes into account the reviewers' comments.

We cannot make any decision about publication until we have seen the revised manuscript and your response to the reviewers' comments. Your revised manuscript is also likely to be sent to reviewers for further evaluation.

Sincerely,

Christian Hilbe

Guest Editor

PLOS Computational Biology

Stefano Allesina

Deputy Editor

PLOS Computational Biology

Guest editor: 

Thank you for your manuscript. In the meanwhile, it has been seen by three experts in evolutionary game theory and dynamical systems. They all agree that the manuscript is sound and that it describes an interesting mechanism for cooperation. Based on my own reading, I fully agree. The reviewers also make a number of very constructive suggestions for how to further improve the paper. Please take all these suggestions into account when preparing a revised version.

Let me briefly mention two further suggestions.

First, it seems to me the present setup is better explained by saying that group members have a choice between different public good games to play, instead of saying that different public resources "compete". I find the term "competition" somewhat misleading. It suggests the resources themselves have strategic motives.

Second, upon my first reading, I had some problems to clearly see which mechanism prevents cooperation from going extinct in this model. Only after closer inspection of Eq. (3), I realized that it's crucial that the present model allows for smaller effective group sizes. That is, even if the baseline group size is g, the actual group in which a player interacts can be considerably smaller. If the effective group size is sufficiently small (i.e., smaller than the respective multiplication factor r_i), cooperation becomes a dominant strategy. All of this is implicit in the payoff Eq. (3), but I would appreciate if the author mentions this aspect more explicitly. Please also note that in Eq. (3), the term with r_i needs to be multiplied by c (note that for the simulations this is inconsequential, because there the parameter c has been set to one).

Reviewer's Responses to Questions

Reviewer #1: This paper deals with the evolution of cooperation in public goods games (PGG) when players can choose between multiple public resources to contribute to. Individuals not only have a (deterministic) strategy for the PGG, but also a preferred game they want to play. In each time step, random groups of a certain size are formed from the population. Players enter their preferred game, interact according to their strategy, and receive the corresponding payoff and an additional base payoff. Then, the population reproduces, subject to mutations. The authors show that the competition between different public goods games can help cooperation evolve. This is due to the fact that cooperators can find shelter and survive in the less profitable resource while defectors predominantly join the higher quality resource. Cooperators can then outcompete the defectors, and this effect is strongest when the public resources have similar quality, i.e. enhancement factors r. Solving the replicator dynamics of this system, the authors find two possible attractors depending on initial conditions and the different qualities of the public resources: either a fixed point or a limit cycle. In the latter case, cooperators are preserved and cyclically dominate the population. Meanwhile, for too small enhancement factors, the dynamics settles in a fixed point where defection dominates. The paper then further explores the system's dependence on the different enhancement factors in more detail, additionally supported by simulations. Robustness results are also provided in the Supplementary Information. The authors come to the conclusion that this mechanism for the evolution of cooperation does not suffer from the decrease in cooperation rate as group size gets larger, which is a contrast to the corresponding observation in classic PGGs.

The paper and its results are interesting, and its motivation clear, which is well explained already in the abstract. Also, the good match between simulations and analytical results inspires confidence. However, I find some issues with this work that should be addressed before publication.

For one, I am a bit confused regarding the definition of the densities of cooperators and defectors, respectively. What is the quantity that these densities are taken relative to? This is never made quite clear, and I somehow would have expected that the density of cooperators and the density of defectors should add up to one, considering there is no other strategy at play? Instead, Fig. 1(b) does not look like this is true, which is puzzling to me. Also, how is the density of cooperators with respect to one game type defined? It is hard to interpret the figures without that information. Furthermore, can one conceptually even add up cooperation in the two (or more) different games to get a global "cooperation density"? I would appreciate some insight on this.

Another question I have is the relation of these results to the results by Hauert et al. in Reference [5], which deals with volunteering/abstaining in the PGG. The authors of that work show that the option to abstain can give rise to cooperation in a rock-paper-scissors cycle, as cooperators start to abstain when defector numbers rise, which in turn enables small groups of loners to switch back to cooperation. Considering that one could see cooperators switching their preferred game is a kind of "abstaining", I would be curious to see a comparison of the two papers in a bit more detail than is currently provided.

Furthermore, it would be important to streamline the paper, such that it flows better, and also remove typos and grammatical mistakes at the same time. E.g., in the results section, crucial statements are repeated multiple times, which certainly reminds the reader of the main message of the paper, but inhibits the reading flow quite a bit. It would be good to give this section a bit more structure. As for typos, e.g. in the author summary, the first sentence is out of order, and in Fig.1, the right y-axis label of panel (b) is off. Also, there is a stray "Lorem ipsum" on page 8 in the paragraph "S2 Fig."

Minor comments:

(-) One pretty important mechanism that gives rise to cooperation in social dilemmas is direct reciprocity (repeated interactions). Why is this not mentioned?

Reviewer #2: In "Competition between public resources promotes cooperation" authors show that competition between public resources provides a simple mechanism for cooperation. In particular, since defectors might be tempted to join the highest quality resource pool, cooperators get a window of opportunity by using lower quality resources. Not surprisingly, this mechanism works best when the qualities of the two highest quality resources are similar, which is also when cooperation is most abundant.

The proposed model is intuitive and clear, and I agree it can be used in many biological and social contexts where individuals can form different groups or join different institutions to perform a collective task. In general, how and why cooperation emerges in the public goods game is an intensely investigated subject with obvious practical ramifications. Methods of computational biology, network science, and statistical physics have been applied successfully and with much effect recently to shed light on the problem from many different perspectives, and also to outline many different ways on how cooperation could be promoted. In this sense, the study certainly addresses a relevant setup, and it also delivers results that might be of interest to the readership of PLOS Computational Biology.

I am in general in favor of publication subject to taking into account the following comments with care and love to detail.

I would advise against explicitly claiming novelty in the abstract and elsewhere. Similar mechanisms have been observed before, so this is hardly the first time we see that competition for limited resources, if presented rightly, can foster cooperation.

One related paper from 2014 is Excessive abundance of common resources deters social responsibility, Sci. Rep. 4, 4161 (2014), which I am sure would fit very well to the introduction, especially related to the importance of the competition the author is focusing on. A more recent paper also focusing on resources in public cooperation is Rewarding endowments lead to a win-win in the evolution of public cooperation and the accumulation of common resources, Chaos, Solitons & Fractals 134, 109694 (2020). A popular review for the public goods game is also Evolutionary dynamics of group interactions on structured populations: A review, J. R. Soc. Interface 10, 20120997 (2013).

I wonder about the robustness of the presented results in structured populations. Can the author please comment on that.

The last section should become a discussion, where not only the results are reiterated, but where also some time in spent on where we could actually expect they might be useful. Here the author can spell out many biological and social contexts where individuals can form different groups or join different institutions to perform a collective task that he mentions in the abstract and introduction.

I also notice some "Lorem ipsum" in the figure captions, which should be double checked and corrected.

References need a careful check for correct punctuation, capitalization, and overall accuracy.

Reviewer #3: This work provides an insight into the emergence of cooperation in the public's good game (in an evolutionary context). More specifically, the author explores the effect that an enhancement factor has on the emergence of cooperation, in a population of cooperators and defectors.

The author considers a population where individuals are divided into groups. Each individual can choose a strategy and a game. Each game is a public’s good game of n players. The difference between the games is the enhancement factor. All the contributions at the end of the round are multiplied by an enhancement factor and are distributed equally among the players.

The author presented results in the case of n=2 and in the case of two different enhancement factors, where commonly one is larger. The results are based on analytical calculations (using the replicator-mutation dynamics equation) and simulations. These show that:

- Defectors tend to play the game with the highest enhancement factor which allows cooperators to survive.

- Cooperation level is maximized when the enhancement factors have the same value.

Furthermore, the author includes some discussion on results for the case of n=3 and an analysis on the robustness of the results for other parameter values.

This work does provide interesting insights to the public's good game. It’s a simple yet interesting formulation that allows the author to show that their mechanism can allow cooperation to occur.

However, it does have some issues, and for this reason I cannot recommend publication in its present form. More specifically:

Material and Methods section.

- The author needs a reference for the replicator-mutation dynamics equation.

- The author needs to add at least a sentence explaining Eq. (1).

- Before explaining Eq. (3) the author states that j is used for the public resources, then produces the explanation using i.

The quality of the graphs.

In Figure 1, the author uses dotted lines and solid lines to distinguish between the density of defectors and cooperators. The size of the sub-figures make it hard to see the different line styles. I propose that Figure 1 should include only subfigures 1a and 1b. Moreover, the author uses color for the axis. If printed in black and white it is not visible. I propose that the author uses a legend instead. Also, Figure 1b, the label of the right y axis is incorrect.

The author uses the line color of Figures 1c and 1d to explain the results in the text. This should be avoided. The reader might have the manuscript in black or white, or simply is colorblind. The figures should be bigger, just have these two sub figures as Figure 2, and use different line styles.

Data and computer code.

The author states that “All data will be made available online after acceptance”. I encourage the author to make the data available and reference them in the paper. Furthermore, I would like to encourage the author to make the computer code for the simulations available as well and reference it in the paper.

Supplementary Information.

In the later parts of the “Results” section the author discusses some further results that have been produced, for example in the case of n=3. These results are currently on the Supplementary Information. I believe that the plots accompanying the results should be included in the Results section.

Literature.

The author discusses previous literature on some mechanisms of the emergence of cooperation, such as reputation effects, network and spatial structure, voluntary participation, reward, and possibly punishment. However, they do not include discussion on using groups or/and different games. I think that a sentence is missing to clarify where this paper fits in the literature.

Minor and other issues.

- Line 2: the author refers to Publics good game with a capital “P”, whereas in the rest of the text the author uses a lowercase “p”.

- Line 49: “as shown in the method section”. Reference the method section properly.

- Line 52: that the dynamics can settle… missing the word “that”.

- The misuse of commas. There are a few points where the comma is not needed. For example:

- Line 38: are formed at random (remove comma) from the population pool.

- Line 40: they receive a based payoff pi_0 (remove comma)

- The misuse/overuse of the word “here”. In several parts of the manuscript the word “here” can be removed or replaced. For example, following Eq 2, in line 181, line 55, line 146 and line 147.

**Have all data underlying the figures and results presented in the manuscript been provided?**

Reviewer #1: **No: **The authors will make the data available after publication.

Reviewer #2: Yes

Reviewer #3: **No: **The author states that the data will be made available after acceptance.

PLOS authors have the option to publish the peer review history of their article (what does this mean?). If published, this will include your full peer review and any attached files.

Reviewer #1: No

Reviewer #2: No

Reviewer #3: **Yes: **Nikoleta E. Glynatsi
---

## [Decision Letter · Decision Letter 1]

29 Dec 2020

Dear Dr. Salahshour,

Thank you very much for submitting your manuscript "Freedom to Choose between Public Resources Promotes Cooperation" for consideration at PLOS Computational Biology. As with all papers reviewed by the journal, your manuscript was reviewed by members of the editorial board and by several independent reviewers. Based on the reviews, we are likely to accept this manuscript for publication, providing that you modify the manuscript according to the review recommendations.

Sincerely,

Christian Hilbe

Guest Editor

PLOS Computational Biology

Stefano Allesina

Deputy Editor

PLOS Computational Biology

[LINK]

Comments by the guest editor: 

The manuscript has been considerably revised. Now all of the previously consulted reviewers suggest publication of the manuscript, subject to minor adjustments. This is also what I would like to suggest.

Please take all the remaining reviewer comments into account when preparing the final version of the manuscript. In addition, please make the following (minor) changes:

(-) I appreciate that the author changed the title of the paper, as I believe it would be misleading to speak of "competing public resources". After all, the resources themselves have no strategic interest in attracting individuals; they are not participating in a competition.

However, even in the revised manuscript, references to competing resources are found throughout the text. Please revisit each of them and consider revising the respective formulation. For example, the last sentence of the abstract currently reads: "... they can perform the most competitively to attract individuals." This could be rewritten as "they are almost interchangeable" or "they approximately attract the same number of individuals." Alternatively, the sentence could just be omitted.

(-) At several places, the author chooses the formulation "Cooperation level is maximized...". This formulation sounds odd to me. Please either use "Cooperation is maximized" or "The cooperation level of the population is maximized"

(-) Page 2, Line 9: "empirical evidence suggests from bacteria to animals and humans..."

-> "empirical evidence from bacteria to animals and humans suggests..."

(-) Page 2, Line 15: Because most readers (me included) will not know what the difference between "reward" and "reward dilemma" is, please consider combining these two items as "reward"

(-) Page 2, Line 42: "and gather payoff" -> "and derive payoffs"

(-) Page 5, Line 146: "As mentioned in the The Model, the model..." ->

"As mentioned in the section The Model, the framework..."

(-) Page 6, Line 187: "In the region where both r1 and r2 are larger [...] it becomes almost obsolete." Please replace "it" by "the third resource"; otherwise the sentence is ambiguous. Same comment for page 7, line 197.

(-) Page 6, "is not too smaller than that of the two higher-quality resources"

-> "is not too small compared to the enhancement factor of the two higher-quality resources"

(-) Page 7, line 227: "appear to abound in different species" -> "appear to be abound in different species"

(-) In general, the author seems to have problems with the correct use of commas. I believe it would be useful if he could ask a colleague for help when preparing the final version of the manuscript.

(-) I would appreciate if already the intro briefly mentions what the main mechanism for cooperation in this manuscript is (When having the choice between multiple public good games, individuals experience smaller effective group sizes. In sufficiently small groups, cooperation is dominant). Otherwise readers will have to wait until the very end (the Discussion) to really understand what is going on.

Reviewer's Responses to Questions

Reviewer #1: The author has satisfactorily taken my comments into account when preparing the revised version of the manuscript. Some very minor issues remain:

(-) The first sentence of the author summary should be “How is selfishness curbed in many biological populations...” instead of “How in many biological populations selfishness is curbed...”.

(-) The caption for Fig. 4 mentions “green squares”. This should be “green circles”.

(-) Section “The case of n=3 resources”: the first sentence contains an errant “the”.

(-) page 8, last sentence of the second to last paragraph: “fate”, not “faith”.

(-) page 8, last paragraph: The author mentioned in the response to the reviewer that preliminary simulations strengthen his assumption that population structure can be beneficial in this setup. Maybe it would be good to very briefly note this, or at least provide a short insight why this assumption seems correct.

Reviewer #2: The authors have revised their manuscript comprehensively and with love to detail. I warmly recommend publication in present form.

Reviewer #3: I would like to open this review by saying that in the point-by-point letter the author thanks each reviewer individually by stating: “I am thankful to the reviewer for his careful reading and assessment of the manuscript”. Gendered language like this should be avoided in the future. The purpose of this is to avoid word choices which may be interpreted as biased, discriminatory or demeaning by implying that one sex or social gender is the norm.

Overall, the author has taken my comments into account when revising the manuscript. The narrative of the paper, the quality of the figures, and the mathematical explanation of the model have improved.

I have some minor issues regarding the discussion which does not read well in its current form. There are some minor language issues, thus I recommend that the author has a final revision before publishing. Some examples, the abbreviation PGG is used though it is not defined, line 252 misses the word “that”.

**Have all data underlying the figures and results presented in the manuscript been provided?**

Reviewer #1: Yes

Reviewer #2: Yes

Reviewer #3: Yes

PLOS authors have the option to publish the peer review history of their article (what does this mean?). If published, this will include your full peer review and any attached files.

Reviewer #1: No

Reviewer #2: No

Reviewer #3: No
---

## [Editor Report · Decision Letter 2]

11 Jan 2021

Dear Dr. Salahshour,

We are pleased to inform you that your manuscript 'Freedom to Choose between Public Resources Promotes Cooperation' has been provisionally accepted for publication in PLOS Computational Biology.

Best regards,

Christian Hilbe

Guest Editor

PLOS Computational Biology

Stefano Allesina

Deputy Editor

PLOS Computational Biology

Response of the guest editor: 

The author has taken all remaining comments into account. The manuscript is now appropriate for publication.

---

## [Editor Report · Acceptance letter]

2 Feb 2021

PCOMPBIOL-D-20-01646R2 

Freedom to Choose between Public Resources Promotes Cooperation

Dear Dr Salahshour,

I am pleased to inform you that your manuscript has been formally accepted for publication in PLOS Computational Biology. Your manuscript is now with our production department and you will be notified of the publication date in due course.

With kind regards,

Alice Ellingham
